

# To trust or not to trust an explanation: using LEAF to evaluate local linear XAI methods

Elvio Amparore[1,2], Alan Perotti[2] and Paolo Bajardi[2]

[1] Department of Computer Science, University of Turin, Turin, Italy
[2] ISI Foundation, Turin, Italy

## ABSTRACT

The main objective of eXplainable Artificial Intelligence (XAI) is to provide effective explanations for black-box classifiers. The existing literature lists many desirable properties for explanations to be useful, but there is a scarce consensus on how to quantitatively evaluate explanations in practice. Moreover, explanations are typically used only to inspect black-box models, and the proactive use of explanations as a decision support is generally overlooked. Among the many approaches to XAI, a widely adopted paradigm is Local Linear Explanations—with LIME and SHAP emerging as state-of-the-art methods. We show that these methods are plagued by many defects including unstable explanations, divergence of actual implementations from the promised theoretical properties, and explanations for the wrong label. This highlights the need to have standard and unbiased evaluation procedures for Local Linear Explanations in the XAI field. In this paper we address the problem of identifying a clear and unambiguous set of metrics for the evaluation of Local Linear Explanations. This set includes both existing and novel metrics defined specifically for this class of explanations. All metrics have been included in an open Python framework, named LEAF. The purpose of LEAF is to provide a reference for end users to evaluate explanations in a standardised and unbiased way, and to guide researchers towards developing improved explainable techniques.

## INTRODUCTION

In recent years, the field of Machine Learning (ML) has experienced a surge in practical applications. However, many ML models, spanning from random forests to deep neural networks, do not provide a human-understandable clarification of their internal decision processes: this issue is known as the *black-box problem* (*Burrell, 2016*).

The lack of explanatory power hampers the deployment of ML in real-world scenarios for a variety of reasons. As a first limitation, black-box algorithms are poor decision support systems (*Doran, Schulz & Besold, 2017*): domain experts such as doctors or financial analysts would hardly take into account an algorithmically-generated second opinion without an argument to support it, or a measurable score of trust (*Jiang et al., 2018*). From a legal viewpoint, in the European Union AI systems are regulated by law with the

Corresponding author
Elvio Amparore,
elviogilberto.amparore@unito.it

General Data Protection Regulation (GDPR)—which includes directives on algorithmic decision-making (*Goodman & Flaxman, 2016*). For instance, GDPR states that: (i) *The decisions which produces legal effects concerning (a citizen) or of similar importance shall not be based on the data revealing sensitive information, for example about ethnic origins, political opinions, sexual orientation, …*; and (ii) *The controller must ensure the right for individuals to obtain further information about the decision of any automated system*. While the required explanations of a complex socio-technical system concern the decision process as a whole, clearly good explainability algorithms would help supporting the provisioning of detailed explanations for ML components (*Edwards & Veale, 2017*).

In recent times, widely used ML models turned out to be biased against racial (*Obermeyer et al., 2019*) or gender (*Hamilton, 2019*) groups. While interpretable-by-design models should be preferred in high-stakes decisions (*Rudin, 2019*), algorithmic fairness is of paramount importance, and its very first step is the explanation of black-box outcomes (*Panigutti et al., 2020*). Furthermore, an unexplained ML model might also be right for the wrong reasons, due to the algorithm learning spurious correlation in the data. Being capable of assessing the reasons why a ML component outputs a given classification is essential to establish trustable AI (*Ignatiev, 2020*).

In this paper we focus on the class of explanation models named *local linear explanations* (LLE). A taxonomy of other approaches can be found in *Belle & Papantonis (2020)*. Explainability methods that explain arbitrary black-box models (*Ribeiro, Singh & Guestrin, 2016a*) using LLEs provide an explanation in terms of the most *relevant features* for a data point classification. The XAI literature considers several evaluation criteria (*Bohanec, Borštnar & Robnik-Šikonja, 2017*; *Guidotti et al., 2018*), but there is generally no consensus on a core of fundamental metrics, and in some cases these properties lack a precise definition (*Gilpin et al., 2018*). Furthermore, the explanation process has to be contextualised with respect the audience for which explainability is sought (*Arrieta et al., 2019*)—for example, the same clinical black-box decision could be explained very differently to patients, doctor, and medical data analysts. Finally, a natural application of explanations consists in supporting a decision making task in order to perform the minimum change on an instance that modifies the classification label. Proactive use of explanations can be found for instance in the credit sector (*Grath et al., 2018*), to guide an applicant in amending his/her position in order to get a loan approval, or in planning (*Fox, Long & Magazzeni, 2017*). To the best of our knowledge, there is no agreement on how to quantify the prescriptive exploitation of local linear explanations.

The main contribution of this paper is a clear and unambiguous definition of a core set of metrics to compare and evaluate explainability methods that explain black-box models with LLEs. We propose to quantify three new LLE aspects: (i) the *local concordance* of the white-box model w.r.t the black-box model for the instance to be explained, under the constraint of explanation conciseness; (ii) the tendency of an explainability method to produce the same explanation on the same data point (named *reiteration similarity*); and (iii) how good the explanation is when it is taken as a recipe to change a data point classification (*prescriptivity*). The core set of metrics also includes existing LLE metrics taken from literature (*local fidelity*). To encourage users to adopt a single, reproducible

definition of these metrics for LLEs, we provide an implementation in the form of a Python framework, named LEAF (*Local Explanation evAluation Framework*). We apply LEAF on the two most used model-agnostic explainability algorithms for LLEs, LIME and SHAP, and show that they display unexpected behaviours that were not previously reported in literature.

The paper is structured as follows: in 'Background' we provide an overview of basic concepts concerning Local Linear Explainers as well as the two techniques we will compare, LIME and SHAP. In 'Methodology' we introduce the LEAF framework and focus on the metrics it computes to evaluate LLE explanations. In 'Results' we apply LEAF on a range of datasets and black-box classifiers, evaluating the explanations provided by LIME and SHAP, and discussing the results. We end the paper with a general discussion ('Discussion') and final considerations ('Conclusions').

## BACKGROUND

In order to tackle the black-box problem, several XAI techniques have been proposed (*Goebel et al., 2018*; *Došilović, Brčić & Hlupić, 2018*), exploiting different principles and processes, usually with supporting graphical representations (*Tamagnini et al., 2017*) and visual analytics (*Choo & Liu, 2018*). In general the goal is to extract human-understandable knowledge from trained black-box models (*Ming, Qu & Bertini, 2018*; *Ribeiro, Singh & Guestrin, 2016b*; *Ribeiro, Singh & Guestrin, 2016c*). However, despite strong consensus about the necessity of enriching ML models with explainability modules, there is no shared definition of what an explanation should be (*Lipton, 2016*; *Pedreschi et al., 2019*), nor quantitative comparisons of different methods (*Murdoch et al., 2019*).

Model-agnostic explanation models are typically based on decision trees, rules or feature importance (*Guidotti et al., 2018*; *Freitas, 2014*; *Craven & Shavlik, 1995*), because of the simplicity of such explanations. Several model-specific and data-specific explanation models have also been developed, e.g., for deep neural networks (*Binder et al., 2016*; *Selvaraju et al., 2019*), deep relational machines (*Srinivasan, Vig & Bain, 2019*), time series (*Karlsson et al., 2019*), multi-labelled and ontology-linked data (*Panigutti, Perotti & Pedreschi, 2020*) or logic problems (*Biecek, 2018*); software toolkits including the implementation of various XAI algorithms have been also introduced (*Arya et al., 2019*). A comprehensive survey of explainability methods can be found in *Guidotti et al. (2018)* and in *Došilović, Brčić & Hlupić (2018)*.

### Local linear explanation methods

In this section we review the class of explanation models named *local linear explanations* (LLE), also called *feature importance* models (*Guidotti et al., 2018*), *additive feature attribution methods* (*Lundberg & Lee, 2017*) or *linear proxy models* (*Gilpin et al., 2018*). Let $x$ be a data point of a dataset $X$, that is fed to a *black-box* model $f$. For the sake of generality, no assumption on the details of $f$ are made. A XAI method explains the value $f(x)$ with a LLE by building an interpretable, *white-box* classifier $g$ that mimics $f$ around $x$.

Formally, let $\mathcal{F}$ be the set of input features of $X$, and $F = |\mathcal{F}|$ be its cardinality. Let $X \subseteq \mathbb{R}^F$ be the input dataset, with $\mu_X$ and $\sigma_X^2$ the vectors of feature means and variances, respectively. For simplicity, we consider $X$ to be a tabular dataset in an *interpretable space*, i.e., the same space where explanations are provided. Let $f : \mathbb{R}^F \to \mathbb{R}$ be the original black-box model, and let $x \in \mathbb{R}^F$ be the input to be explained. We consider white-box models in the form of *local linear explanations* (LLE).

A LLE model $g$ is a linear function of the input variables of the form

$$g(x) = w_0 + \sum_{i=1}^{F} w_i \cdot x_i \tag{1}$$

$g$ assigns to each feature $i$ a weight $w_i$, in order to approximate the behaviour of $f$ in the local neighbourhood of $x$. Intuitively, the absolute value $w_i$ of each feature $i$ gives its *importance* in the explanation. In some cases, only $K \leq F$ features have non-zero weights.

In the following subsection we will describe LIME and SHAP in detail. Several other explainability methods (*Guidotti et al., 2018*) for specific classes of black-box models (i.e., model-aware) exist. For instance, DeepLIFT (*Shrikumar, Greenside & Kundaje, 2017*) is a recursive prediction explanation for deep learning models, which introduced the idea of *background values* also used by SHAP. Another approach, LACE (*Pastor & Baralis, 2019*), combines the local perturbation concept of LIME together with the computation of *Shapley*-like values used in SHAP. Other approaches tackle explainability from a different perspective, i.e., they do not provide explicit indications about the reasons why a decision has been taken but rather generate counterfactual explanations through optimisation-based methods (*Ustun, Spangher & Liu, 2018*; *Karimi et al., 2019*; *Grath et al., 2018*).

## LIME: local interpretable model-agnostic explanations

The LIME method (*Ribeiro, Singh & Guestrin, 2016c*) constructs LLE models $g$ starting from a synthetic neighbourhood $N(x)$ around the input to be explained $x$. A *local neighbourhood* $N(x)$ of $H$ points around an instance $x$ is defined as

$$N(x) = \left\{ x_j = x + p_j, \ p_j \sim \mathcal{N}(0, \sigma_X) \ \middle| \ j = 1 \ldots H \right\} \tag{2}$$

where each vector $p_j$ represents a local perturbation, $\mathcal{N}$ is the multivariate normal distribution, and $\sigma_X$ is the vector of feature variances in the training dataset. For simplicity, we do not consider how to randomly generate categorical features, which can be uniformly sampled from their frequency in the input dataset $X$. To find $g$, LIME fits a ridge regression model to $N(x)$ using the linear least squares function

$$\mathcal{L}(f, g, \pi_x) = \sum_{z \in N(x)} \pi_x(z)(f(z) - g(\eta(z)))^2$$

where the default distance kernel $\pi_x(z) = \exp\left(-d(x, z)^2 / \gamma^2\right)$ is a weighted distance function for a kernel width $\gamma = \frac{3}{4}\sqrt{F}$, and $d(\cdot, \cdot)$ is the Euclidean distance. Explanations can be given in an explanation space $X'$ which can differ from $X$, provided that a mapping function $\eta : X \to X'$ is given ($\eta$ is the identity when $X \equiv X'$).

LIME is designed to select a subset of the $F$ features for the LLE model $g$ iteratively, by fitting multiple ridge regressors using a *sequential feature selection* (SFS) algorithm

(*Pudil, Novovičá & Kittler, 1994*). Intuitively, the resulting model $g$ is a *local explanation* for $x$ because it is built to classify the behaviour of $N(x)$, which is a synthetic dataset built around $x$. It is a linear explanation because it provides a single scalar weight for each feature of $x$. A different distance kernel $\pi_x(z)$ can be used, altering the resulting explanation.

## SHAP: shapley additive explanations

The SHAP method (*Lundberg & Lee, 2017*) (also called IME, Interactions-based Method for Explanation) derives local explanation models using the concept of *Shapley values* from cooperative game theory (*Moretti & Patrone, 2008*; *Moeyersoms et al., 2016*). In principle, the Shapley theory applies to binary features only, but it can be extended to real values. Many variations of SHAP have been defined: in this paper we will refer to the *KernelExplainer*, which is the model-agnostic and most general version of the method. Other model-specific variations, like *TreeExplainer* for tree-based models (*Lundberg et al., 2020*), have also been defined, but are not considered in this paper due to their model-aware nature.

A SHAP explanation is a vector $\phi = (\phi_0, \phi_1 \ldots \phi_F)$ that assigns a feature importance $\phi_i$ to each input feature. Intuitively, the input features of a classifier are akin to players cooperating to win a game (the model prediction). The more important a player $i$ is to the cooperation, the higher is its Shapley value $\phi(i)$. Features are grouped into *coalitional sets*, corresponding to the power set of the set of features $F$.

To evaluate the black-box model for a subset of features $S \subseteq \mathbb{F}$ (i.e., the coalition with only players from $S$), SHAP introduces the concept of *background values* $\mathcal{B}$, to replace the missing features in the evaluation of $f$. Let $f_S(x_S)$ denote the evaluation of $f$ on $x$ where only the features in the subset $S$ are kept, and the others are replaced with the background values from $\mathcal{B}$. The set $\mathcal{B}$ is usually taken as either a single sample $\mathcal{B} = \{\mathbb{E}(X)\}$ being the dataset average, or as a set of centroids of $X$.

For a feature $i \in \mathcal{F}$, its Shapley value $\phi(i)$ is defined as follows

$$\phi(i) = \sum_{S \subseteq \mathcal{F} \setminus \{i\}} \frac{|S|! \cdot (F\text{-}|S|\text{-}1)!}{F!} \left( f_{S \cup \{i\}}(x_{S \cup \{i\}}) - f_S(x_S) \right) \tag{3}$$

Let $\phi_0 = f_\varnothing(x_\varnothing)$ be the constant neutral value, where all feature values of $x$ are ignored.

We can derive a LLE model $g$ as a linear regressor using Eq. (1) with weights

$$w_0 = \phi_0, \qquad w_i = \frac{\phi_i}{x_i - \mu_i}, \qquad 1 \le i \le F, \mu_i \in \mathbb{B}_{\{i\}} \tag{4}$$

Such derivation guarantees to be locally accurate (*Lundberg & Lee, 2017*, page 4), i.e., $f(x) = g(x)$. Unlike LIME, SHAP does not build a local neighbourhood $N(x)$ to generate the explanation, relying instead on the computation of the feature importances using Eq. (3).

An important observation is that KernelExplainer of SHAP is deterministic only in its theoretical formulation (*Lundberg & Lee, 2017*; *Štrumbelj & Kononenko, 2010*), but may significantly diverge from such promise in practice, due to implementative details. The exact computation of Shapley values requires to evaluate Eq. (3) on all subsets of $\mathbb{F}$, resuiting in $2^F$ evaluations. However, the official SHAP implementation limits the evaluation to a user-defined number of subsets (default $2F + 2^{11}$). Therefore, when explaining data

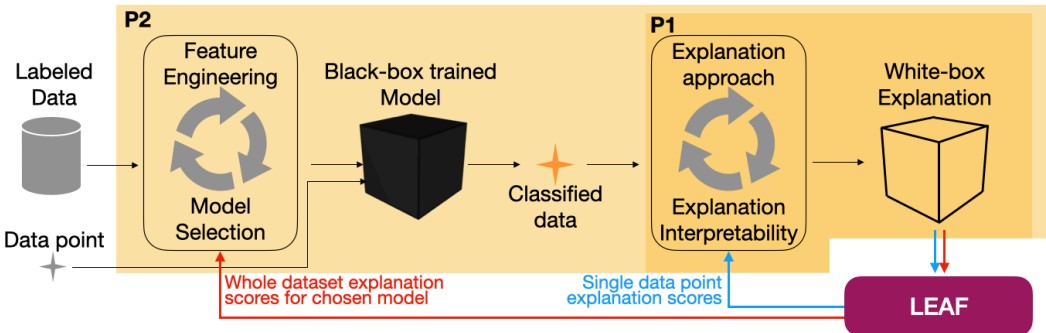

**Figure 1  How LEAF can support the development of a supervised explainable machine learning pipeline.** The standard steps for a supervised machine learning pipeline are: given a labeled data set, features are engineered and models are tested to provide the best prediction accuracy. Once the best model (that might also be an intrinsically black-box model) is identified, it can be used to take decisions and classify previously unseen data points (black arrows). Here, post-hoc explanation techniques can be used to understand the black-box decisions by means of white-box explanations. The LEAF framework can induce two feedback loops in the whole pipeline, both to quantify the goodness and choose the more suitable explanation approach (P1, blue arrows), or to sort out which model is providing the most accurate post-hoc explanations, according to the explanation metrics (P2, red arrows).

points with more than 11 features, SHAP resorts to the Adaptive Sampling (*Štrumbelj & Kononenko, 2014*) heuristic, which approximates the Shapley values using a reduced set of subsets.

## On feature importance

While both LIME and SHAP are based on the concept of feature importance, the definition of what constitutes an "important" feature is not unique. A survey of the different interpretations of the importance concept can be found in *Shrikumar, Greenside & Kundaje (2017)*. LIME adopts the concept of *local importance*. A feature is locally important if its variation around $x$ produces a big change in the output value of $f$. Therefore, a feature that is highly relevant for the classification of $x$, but whose perturbation in the close neighbourhood $N(x)$ does not impact much on the resulting classification, can easily get a low importance. For SHAP, importance is determined in comparison against the *background values* $\mathcal{B}$. A feature is important if its deviation from the background value to $x$ produces a large variation in the output. As a result, LIME and SHAP produce explanations that are not directly comparable and therefore it is of high importance to determine a set of evaluation metrics that are not biased by the different concepts of importance.

## METHODOLOGY

LEAF is a Python framework designed to support several steps of a machine learning pipeline where local linear explanations are part of the final decision process. It is designed around two major perspectives reflecting the most common scenarios where explainability methods might be present, as sketched in Fig. 1, i.e., final decision process and the model development:

- **P1**: *Explaining single decisions.* The black-box classifier $f$ is given. The goal is to understand if an explanation $g$ computed by an explainer on a single instance $x$ can be trusted, and how complex $g$ should be. LEAF supports this use case by providing metrics related to the quality of the explanation, given that the LLE method and a level of explanation complexity are defined. This allows an end-user to trust the white-box explanation, or to reject it if it does not meet the expected quality.

- **P2**: *Model development.* The black-box model $f$ can be chosen to maximise both classification performances and model interpretability. In this case, it might be important to know which classifier should be used to provide, on average, the best explanations. LEAF supports this use case with a feedback loop allowing the comparison of different black-box models in terms of quality of the explanations that can be extracted.

In literature there is no consensus upon nor a unique definition of "explanation quality". However, different aspects of a LLE model $g$ can be measured by different metrics. LEAF includes the most commonly used ones (conciseness, local fidelity), along with new metrics (local concordance, reiteration similarity, prescriptivity) described hereafter.

## Conciseness

We agree with *Guidotti et al. (2018)* that the general definition of interpretability as *to which extent the model and/or the prediction are human-understandable* is pivotal in the evaluation of an explanation; alas, interpretability is often only vaguely defined in literature as correlated to human comprehensibility (*Freitas, 2014*). Due to our quantitative approach, we decided to focus on a specific measurable aspect of interpretability: for LLE models, we introduce the *conciseness* metric as corresponding to the maximum number $K$ of non-zero weights $w_i$ that are kept in the explanation presented to the user, while the other $F - K$ features are treated as non-relevant and are excluded. For datasets with a large number of features, a compact explanation is clearly preferable, for a human reader, over one encompassing all features in the dataset.

LIME and SHAP have different approaches to the conciseness of their explanations. In LIME, the conciseness is given a priori, i.e., the number $K$ is fixed by the user and the explanation is provided with exactly $K$ features, with no guarantee if $K$ is too high or too low. SHAP instead has no direct concept of conciseness, and will always produce a feature importance $\phi_i$ for every feature $F$. However, since the values of $\phi$ already represent a ranking of the features importance, the user could keep the $K$-topmost features, and drop the others to obtain a concise explanation (*Fryer, Strmke & Nguyen, 2021*).

## Local fidelity

It measures how good is the white-box $g$ in approximating the behaviour of the black-box $f$ for the target sample $x$ around its synthetic neighbourhood $N(x)$. The fidelity score can be defined either as a *global fidelity* (see *Guidotti et al., 2018*) or as a *local fidelity* (see *Ribeiro, Singh & Guestrin (2016c)*); hereafter, we will refer to the latter. The agreement between $f$ and $g$ is usually measured using the F1 score (*Guidotti et al. (2018)*, pp. 92). Being a local metric, each sample $x$ will result in a different local fidelity score. By using the neighbourhood $N(x)$ instead of $x$, local fidelity gives an indication of how $g$ behaves in the

locality of $x$, but as a consequence local fidelity is highly dependent on how the $N(x)$ points are sampled. While there is no canonical way to define such sampling (*Andrews, Diederich & Tickle, 1995*), we adopt the one provided in *Ribeiro, Singh & Guestrin (2016c)*, which is the same as Eq. (2). It is worth noting that this definition of local fidelity, albeit common, favours *local importance* and could therefore be biased toward LIME-like methods. This happens because LIME fits the white-box model $g$ minimizing the classification loss for a neighborhood $N(x)$, thus local fidelity captures the optimization target of LIME.

## Local concordance

It measures how good $g$ is in mimicking $f$ for the sole instance $x$ under the conciseness constraint. Local concordance is defined as: $\ell(|f(x) - g(x)|)$, where $\ell(k) = \max(0, 1 - k)$ is the hinge loss function (*Rosasco et al., 2004*), so that the score ranges from 0 for total disagreement, to 1 for a perfect match. In literature there are several similar metrics with different names, like *completeness* (*Sundararajan, Taly & Yan, 2017*), *summation-to-delta* (*Shrikumar, Greenside & Kundaje, 2017*), or the axiomatic definition of *local accuracy* defined in (*Lundberg & Lee, 2017*). However, all these definitions overlook the relationship with conciseness. An explanation is typically pruned to encompass a reduced set $K$ of features. We clarify this ambiguity by evaluating the local concordance for white-box models with only the $K$ requested features. Local concordance is different from local fidelity because it also considers the conciseness constraint, i.e., only the $K$ requested features are considered.

SHAP claims (*Lundberg & Lee, 2017*) the local concordance of $g$ to be always 1. Unfortunately, this is true only in the trivial case when $K = F$, i.e., when all features belong to the explanation. When a conciseness constraint requires $K$ to be less than $F$, deterministic methods like SHAP may also fail to provide a local concordance of 1. We shall provide a detailed analysis of this behaviour in 'Results'. LIME provides no guarantees about the local concordance of its LLE models. Since for both techniques $f(x)$ and $g(x)$ are not guaranteed to be exactly the same, the label of the explanation could be inconsistent with the label predicted by the black-box model.

## Reiteration similarity

Most available explainability methods use some form of randomised/Monte Carlo algorithm to build their explanations. Therefore, if a method is applied several times to explain the same instance $x$, the resulting explanations may vary. The work in *Lipton (2016)* suggests that XAI methods may not converge to a unique solution, while the work in *Zhang et al. (2019)* studies the source of uncertainty of LIME explanations, without providing a general way to quantify LLE stability. As already mentioned in 'SHAP: shapley additive explanations', SHAP defines a maximum number of evaluations of $f$. For small datasets SHAP behaves deterministically; conversely, if $2^F$ exceeds this threshold value, SHAP relies on *Adaptive Sampling* and therefore behaves stochastically. In this case, Shapley values are approximated and SHAP explanations are not perfectly stable.

We propose to measure the similarity of a set of explanations of a single instance $x$ as a measure of similarity across multiple reiterations of the explanation process. Given

two explanations $g, g'$, let $J(g, g')$ be the *Jaccard similarity* between the sets $\Phi(g)$ and $\Phi(g')$ of non-zero weight indices of $g$ and $g'$. Given a set of $R$ explanations of $x$, namely $G_x = \{g_i | i = 1 \dots R\}$, we define the *reiteration similarity metric* as $\mathbb{E}[J(g, g')], \forall g, g' \in G_x$. Note that this definition only considers the selected features, not their weight in the LLE model. A fine-tuned definition that includes also the feature weights and/or the feature rankings (*Webber, Moffat & Zobel, 2010*) could be designed, but it is not considered in this paper.

## Prescriptivity

Most literature in the field deals with the subject of building explanations, without considering how to use such explanation models to take decisions. One intuitive information that an explanation should provide is: "since $x$ is classified as class $A$, what are the minimum changes on $x$ according to the explanation $g$ that lead to a new instance $x'$ that belongs to class $B$?" While there are several examples of such usage of explanations (*Grath et al., 2018*; *Joshi et al., 2019*; *Karimi et al., 2019*; *Ustun, Spangher & Liu, 2018*; *Fernandez, Provost & Han, 2020*), explanators like LIME and SHAP are not designed to explicitly support this kind of use for the explanations they produce. Therefore it is of interest to understand if local linear explanations are applicable in a proactive scenario. We introduce the novel metric of *prescriptivity* to measure how effective a LLE is when taken as a recipe to change the class of $x$. Without loss of generality, we will consider the target classification boundary $y'$ to be $\frac{1}{2}$, but $y'$ could assume any value.

Let $\mathcal{D}_g(y') = \{x \in \mathbb{R}^F | g(x) = y'\}$ be the set of points in the domain of $g$ whose codomain is the constant $y'$. The *boundary* of $g$ is then the set $\mathcal{D}_g(\frac{1}{2})$. Given $x$ and the LLE model $g$, let $x'$ be the projection of $x$ on the boundary of $g$, such that $d(x, x')$ is minimal. The intuition is that $x'$ is the closest point to $x$ that switches its classification, according to the explanation model $g$. Let $h = x' - x$ be the difference vector. Since $g(x) = w_0 + \sum_{i=1}^{F} w_i \cdot x_i$, the point $x'$ is the solution to the equation

$$w_0 + \sum_{i=1}^{F} (x_i + h_i) \cdot w_i = \tfrac{1}{2} \qquad \Rightarrow \qquad h_i = \left(\tfrac{1}{2} - g(x)\right) \cdot w_i^{-1} \tag{5}$$

Figure 2 shows how $x'$ is found on the LLE model $g$ that gives the local explanation of $x$, and it is loosely based on a similar picture in *Ribeiro, Singh & Guestrin (2016c)*. The projection $x'$ of $x$ on the boundary $\mathcal{D}_g(\frac{1}{2})$ is the closest point to $x$ where $g(x') = \frac{1}{2}$. Of course, the value of $f(x')$ could be different from $\frac{1}{2}$, depending on how good $g$ is in approximating the classification boundary.

We define the *prescriptivity* metric as: $\ell\left(\frac{1}{C} \cdot |f(x') - g(x')|\right)$, where $\ell(\cdot)$ is the hinge loss function, and $C = \max(y', 1 - y')$ is a normalisation factor, so that 1 means that $x'$ lies on the boundary, and 0 means $x'$ is at the highest distance from the boundary. Observe that by taking the absolute value, we measure both over-shoots and under-shoots of the boundary as a loss of prescriptivity.

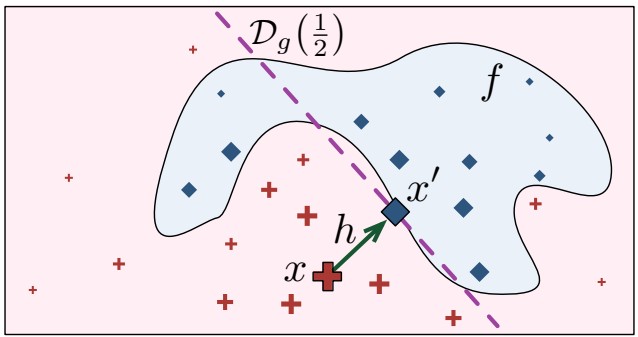

**Figure 2** Finding the prescriptive point $x'$ on the LLE boundary $\mathcal{D}_g\left(\frac{1}{2}\right)$.

## RESULTS

Many reasons might contribute to the popularity of LIME and SHAP: their ability to explain a wide range of black-boxes and different kinds of data (such as images and text), a user-friendly Python implementation that includes helpful visualisations, the adoption and inclusion into some companies XAI solutions are some of them. As a matter of fact, they currently represent the standard approach to tackle explainability in machine learning pipelines. In both methods, however, complex data is always mapped into an interpretable representation in *tabular form* for the explanation. For instance, images could be either partitioned into a bit-vector of *superpixel/pixel* patches, or the pixels could be directly mapped into tabular columns; texts could be converted into a bit-vector of word occurrences. Explanations are then produced in terms of the features (i.e., the columns) of these tabular representations. Therefore, without loss of generality, our experiments focus on the explanations of tabular data, which is easier to analyse and understand, without any intermediate conversion. Moreover, experiments cover binary classification tasks only, as it is common practice among LLE methods (including LIME and SHAP) to map $n$-ary classification using $n$ one-vs-all binary classifiers.

We start by considering the problem **P1** defined in 'Methodology' and addressed it with LEAF key metrics. Given an instance $x$ to be explained, for each explainability method considered, LEAF generates $R$ explanations of $x$. It then shows a summary of the explanation metrics along with an explanation, to help the user understand if the local model $g$ can be trusted.

As an example, we consider the *drug consumption* dataset (*Fehrman et al., 2015*), where the classifier tries to predict whether a patient is inclined to be a frequent consumer of a specific drug (caffeine in this case) based on personality traits ($F = 10$ features). Figure 3 shows how LEAF works on a data point $x$, producing $R = 50$ explanations to compute the reiteration similarity score. The explained black-box classifier $f$ is a *random forest* with $E = 50$ estimators and max. depth of 5. Both the LIME and SHAP methods are evaluated, using 5000 neighbourhood samples for LIME (default), and 5,000 samples to compute the Shapley values for SHAP. While the number of points used by the two tools refers to rather different internal details, we have chosen to take the same number as a basic fair principle.

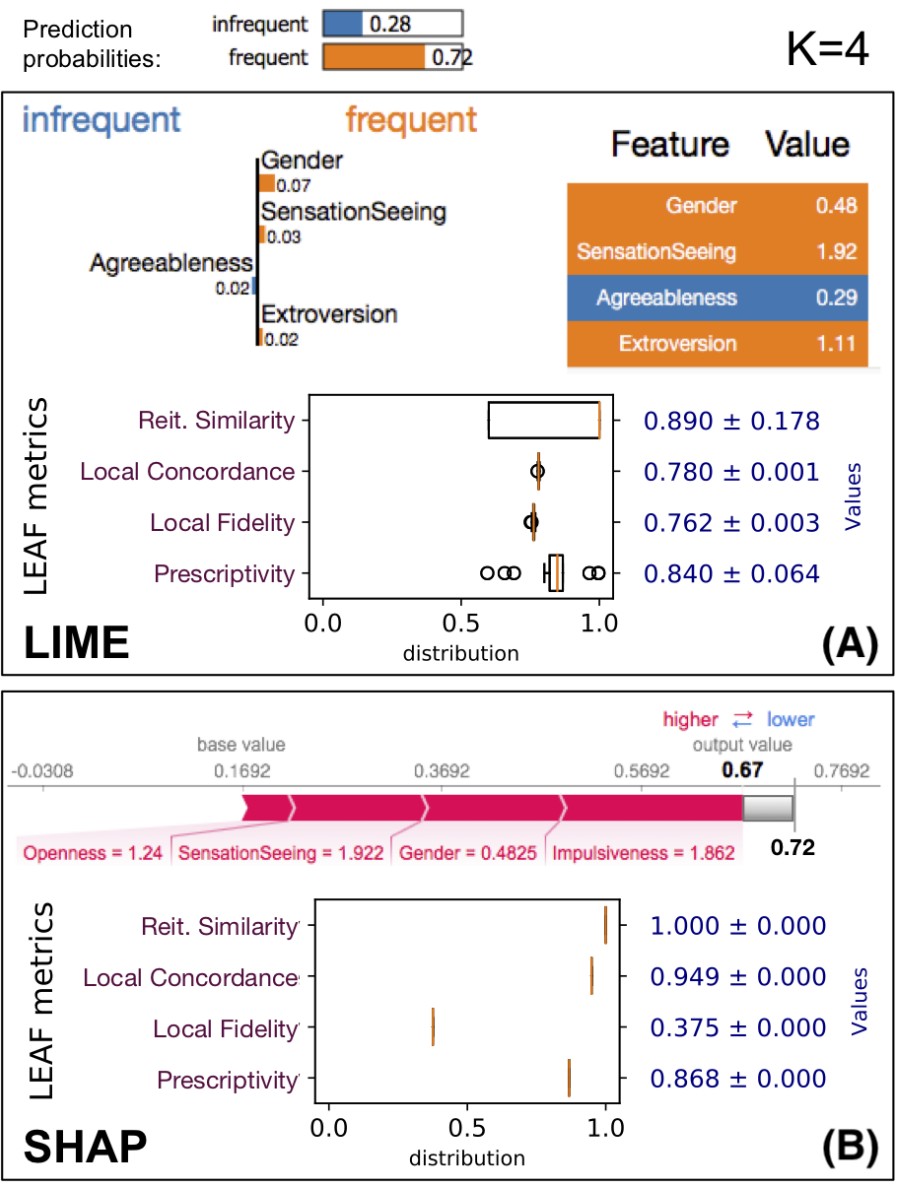

**Figure 3** **LEAF applied to evaluate the quality of the explanations provided by LIME and SHAP on the same data point x, using the same ML model f.** The black-box prediction $f(x)$ is 0.72 for the *frequent* class. (A) The output for LIME with $K = 4$, followed by the four LEAF metrics. (B) SHAP output restricted to the $K = 4$ most relevant features, whose output values sums up to 0.67 instead of 0.72 due to truncation.

Each box in Fig. 3 shows the explainer outputs (top), followed by the summary of the explanation metrics. The summary shows the boxplot of the metric values, as well as their averages. Only the first explanation is shown, even though LIME/SHAP are run $R = 50$ times to obtain 50 explanations so that the metrics, such as reiteration similarity, can be computed. Asking for $K = 4$ most relevant features, LIME and SHAP do not agree on which features are the most important, which is not unexpected given their different definition of

feature importance (see 'On Feature Importance'). These two explanations have different local fidelity and prescriptivity scores. In addition, LIME shows some instability in terms of reiteration similarity, i.e., multiple invocations of the explainer will not result in the same set of relevant features. Moreover, LIME explanations shows an average local concordance of 0.78, which means that $g(x)$ is not very close to the value of $f(x)$. SHAP on the same instance is more consistent, reporting the same explanation over and over. As previously stated, SHAP guarantees a perfect local concordance only when considering all the $F$ features in the LLE, not just the $K < F$ most relevant. In this case the local concordance is 0.949 which is still high. However, the local fidelity of the LLE models of SHAP in the neighbourhood of $x$ is much smaller than the one obtained by LIME explanations. In contrast, SHAP shows to have a better prescriptivity than LIME, i.e., manipulating the top four features identified by SHAP as explanations produces a new point $x'$ with a higher chance of changing the classifier outcome.

Even such a trivial example allows us to gain several insights about the trustability of explanations. First, LIME and SHAP focus on different sets of features. Second, LIME is not very stable, so the trustability of a single explanation is limited. Third, SHAP is perfectly stable and overall achieves better scores than LIME, even if the low local fidelity is something to be considered.

It is worth noting that the explanation local fidelity for this sample (in particular for the explanation provided by SHAP) is not very good (it is 0.762 for LIME and only 0.375 for SHAP). However, when we actually follow the indication of the explanation to reach the new boundary $f(x')$, we see that it is quite close. This could happen because the local fidelity evaluates two different aspects at once: (1) it could be low because the white box model $g$ is not a good model for the neighborhood $N(x)$; (2) the neighborhood $N(x)$ lies close to a non-linear classification boundary, which is not fully captured by $g$. This example shows that the widely-used local fidelity score does not capture the prescriptive use of an explanation, and it is limited in the local evaluation of the white-box model.

This example shows how the metrics provided by LEAF allow a domain expert to take decisions about the trustworthiness of an explainer for the problem under scrutiny.

## Evaluating the reiteration similarity of XAI methods

To be trusted, an explanation needs to be stable, i.e., the explainability method should not provide entirely different sets of relevant features $\Phi(g)$ if called multiple times to explain the same instance $x$. Reiteration similarity is therefore a precondition that needs to be verified. This is even more evident when considering the GDPR remark about the right for individuals to obtain information about the decision of any automated system: clearly, if an algorithmic decision-support system provides inconsistent explanations for the same data point, the single explanation provided to the user cannot be trusted.

Figure 4 shows the *reiteration similarity* metric distribution measured on 100 instances, computed for both LIME and SHAP, for four datasets and six classifier categories (lin, log, rf, kn, mlp, svc), also reported in the legend. Linear classifiers are transparent by design and do not need additional explanations—we have included one as a baseline. Each boxplot shows the reiteration similarity distribution of the explainer on a classifier $f$ on

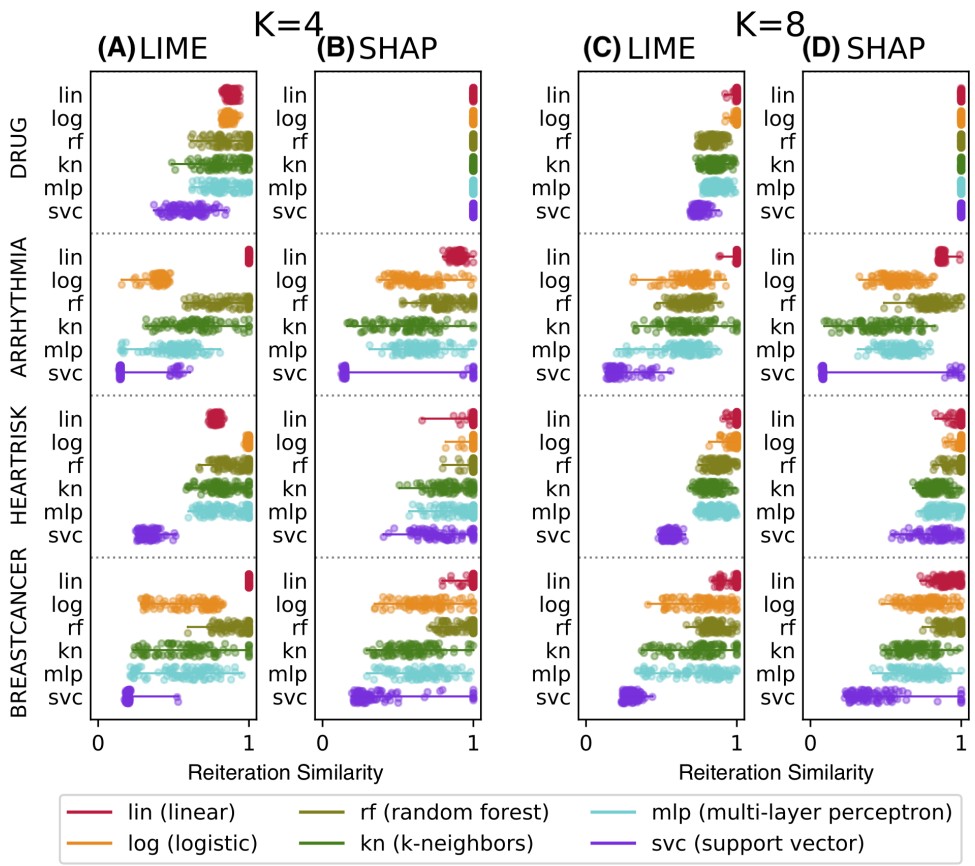

**Figure 4  Reiteration similarity metric on four datasets and six classifier categories, for two conciseness levels $K = 4$ and 8.**

a dataset, for a fixed value of $K$ (4 and 8 on the left and right, respectively). Each of the 100 values is also shown as a small dot, and is again the result of $R = 50$ explanations for each instance, thus resulting in $100 \times 50$ explanations per boxplot. The datasets are: *drug* consumption, *arrhythmia* (*Dua & Graff, 2017*), *heartrisk* (*Doddamani, 2018*), and the *breast cancer* Wisconsin dataset (*Dua & Graff, 2017*), with $F = 10$, 279, 15 and 30 features, respectively.

We used *scikit-learn* (https://scikit-learn.org/) classifiers to test the LEAF framework. The classifiers used in the tests are: *lin* (simple linear classifier); *log* (logistic classifier with *liblinear* solver); *rf* (random forest classifier with 50 estimators, 5 levels max depth); *kn* (k-neighbors with 3 neighbors); *mlp* (simple neural network with 1 hidden layer and 100 neurons); *svc* (support vector classifier with *rbf* kernel and $\gamma = 2$). Table 1 summarises the used datasets and the out-of-samples accuracy reached by the tested classifiers. The train/test ratio used to compute the accuracies is 80/20.

The data shows that the considered methods may produce unstable explanations on some pathological instances or for some classifier categories. The classifiers mlp and svc appear to be the hardest ones to explain. The difficulty of explaining ANN models was

**Table 1  Summary of the datasets used in the experimental section.**

| Dataset | $F$ | Out-of-sample accuracy | | | | | |
|---|---|---|---|---|---|---|---|
| | | lin | log | rf | kn | mlp | svc |
| Drug | 10 | 0.583 | 0.588 | 0.788 | 0.828 | 0.904 | 0.925 |
| Arrhythmia | 279 | 0.635 | 0.746 | 0.838 | 0.684 | 0.765 | 0.500 |
| Heart risk | 15 | 0.665 | 0.670 | 0.740 | 0.832 | 0.865 | 0.892 |
| Breast cancer | 30 | 0.969 | 0.986 | 0.934 | 0.986 | 0.979 | 0.629 |

also noted in *Štrumbelj & Kononenko (2010)*, as well as for svc in *Caragea, Cook & Honavar (2001)*. Nevertheless this remains a bit surprising, since explainability methods have been promoted to explain ANNs and to be model-agnostic in general. SHAP appears to have slightly higher reiteration similarity values on average than LIME, but pathological cases are observed with both approaches.

To further evaluate the impact of non-linearity to the reiteration similarity, we consider in Fig. 5 the same metric on the *heartrisk* dataset for multiple neural network architectures. The plot shows the distribution of the reiteration similarity metric on 500 data points. Data points are split into correctly and incorrectly classified (i.e., $f(x) = y$ or $f(x) \neq y$), to further investigate whether the original data point classification is relevant for the observed instability. Reiteration similarity seems to decline with the increasing non-linearity of the classifier. However, the relation between the non-linearity (both in terms of neurons and hidden layers) and the reiteration similarity score follows a non-trivial pattern. In fact even simpler models may experience instabilities in the explanations over multiple reiterations. We have decided to separately investigate correctly and incorrectly classified data points with respect to their reiteration similarity; empirical observations show that in this case the correctness of the sample classification does not seem to be a major factor for the reiteration similarity scores.

## Evaluating the prescriptivity of explanations

Figure 6 shows an example of using an explanation (generated by SHAP) of an instance $x$ to identify a new synthetic instance $x'$ which is closer to the classification boundary than $x$. This example uses the *heartrisk* dataset (*Doddamani, 2018*), which associates heart risks with personal health traits, and we use a mlp regressor for $f$ with 100 neurons in the hidden layer and *ReLU* activation. The example focuses on a fictional patient: female, aged 49, smoker and with the other health attributes reported in the feature table in Fig. 6(B), column $x$. The SHAP explanation on Fig. 6(A) shows the heart risk of $x$. LEAF uses the selected features of SHAP to generate $x'$ using Eq. (5), which is then explained again by SHAP (bottom of (B)). The initial *at risk* classification ($f(x) = 0.81$) is driven below the *risk* boundary with $f(x') \approx 0.49$ by changing the $K = 3$ features (systolic blood pressure from 235 to 125.5, etc...). A prescriptivity close to 1 indicates that the boundary $\mathcal{D}_g(\frac{1}{2})$ identified by the LLE model $g$ is a reliable indicator of the boundary position.

For this example, we have selected an instance with high prescriptivity, showing how the explanation can be trusted proactively. Since this is not always the case, it is important to quantify whether the explanation can be used in a prescriptive way.

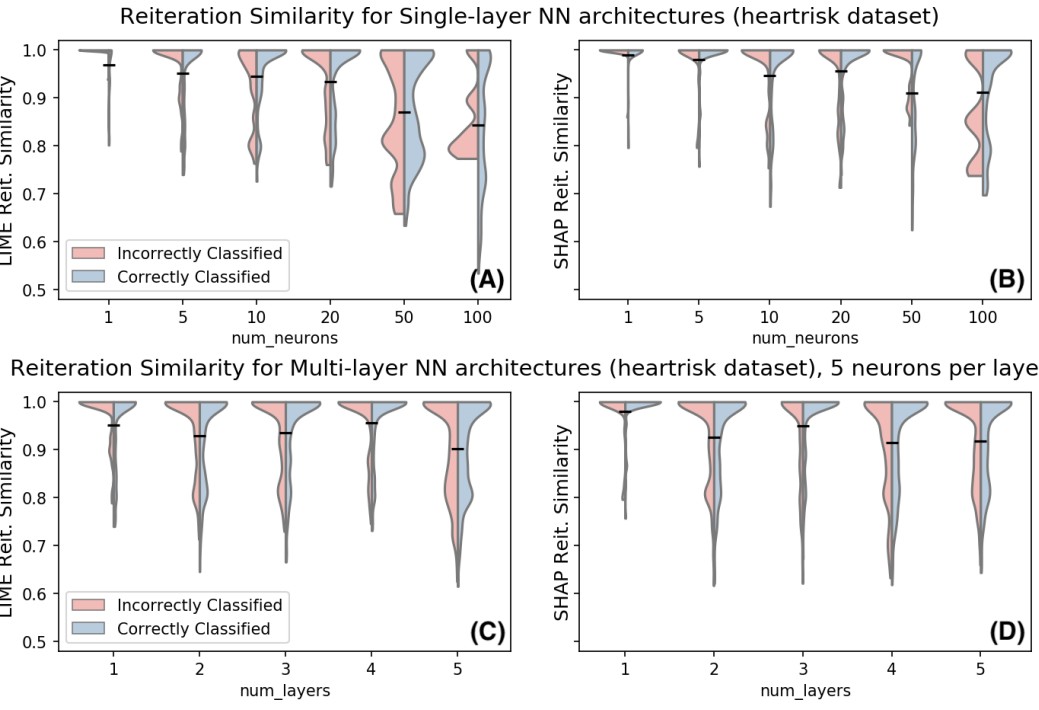

**Figure 5** **Reiteration similarity for LIME (left) and SHAP (right) for multiple Neural Network architectures on the Heartrisk dataset, with K = 4.** (A–B): single layer architecture, the number of neurons in the single hidden layer changes. Out-of-sample accuracies: 0.677, 0.687, 0.685, 0.745, 0.804 and 0.865. (C–D): multiple layers with five neurons each. Out-of-sample accuracies: 0.687, 0.695, 0.709, 0.705 and 0.720.

## Selecting an explainable classifier

We now consider the problem **P2** defined in 'Methodology'. If a classifier may be chosen among different options, LEAF can be used to identify the one that will provide the best results with a XAI method, and the lowest complexity $K$.

As example we consider again the *drug consumption* dataset. Instead of evaluating a single instance for a fixed classifier/conciseness value $K$, in the **P2** scenario we consider six classifiers and three values for $K$. Figure 7 shows the distributions of the evaluation metrics for a test set of 100 instances, each tested $R = 50$ times to compute the reiteration similarity.

Each block in Fig. 7 shows an evaluation metric, computed for different values of $K$ (on the rows). Each plot shows on the left (A, C, E, I, G, K) the values for LIME, and on the right the values for SHAP (B, D, F, H, J, L). Each method is tested for six different classifier categories. Boxplots shows the results of the metric in the given setting, for the 100 tested instances.

This plot can be used to support several decision aspects of **P2**:

• *What explainability method should be selected in order to have stable explanations?* LIME is highly stable only for linear/logistic classifiers, and for lower conciseness levels (higher $K$ values). SHAP instead shows excellent reiteration similarity even at low conciseness.

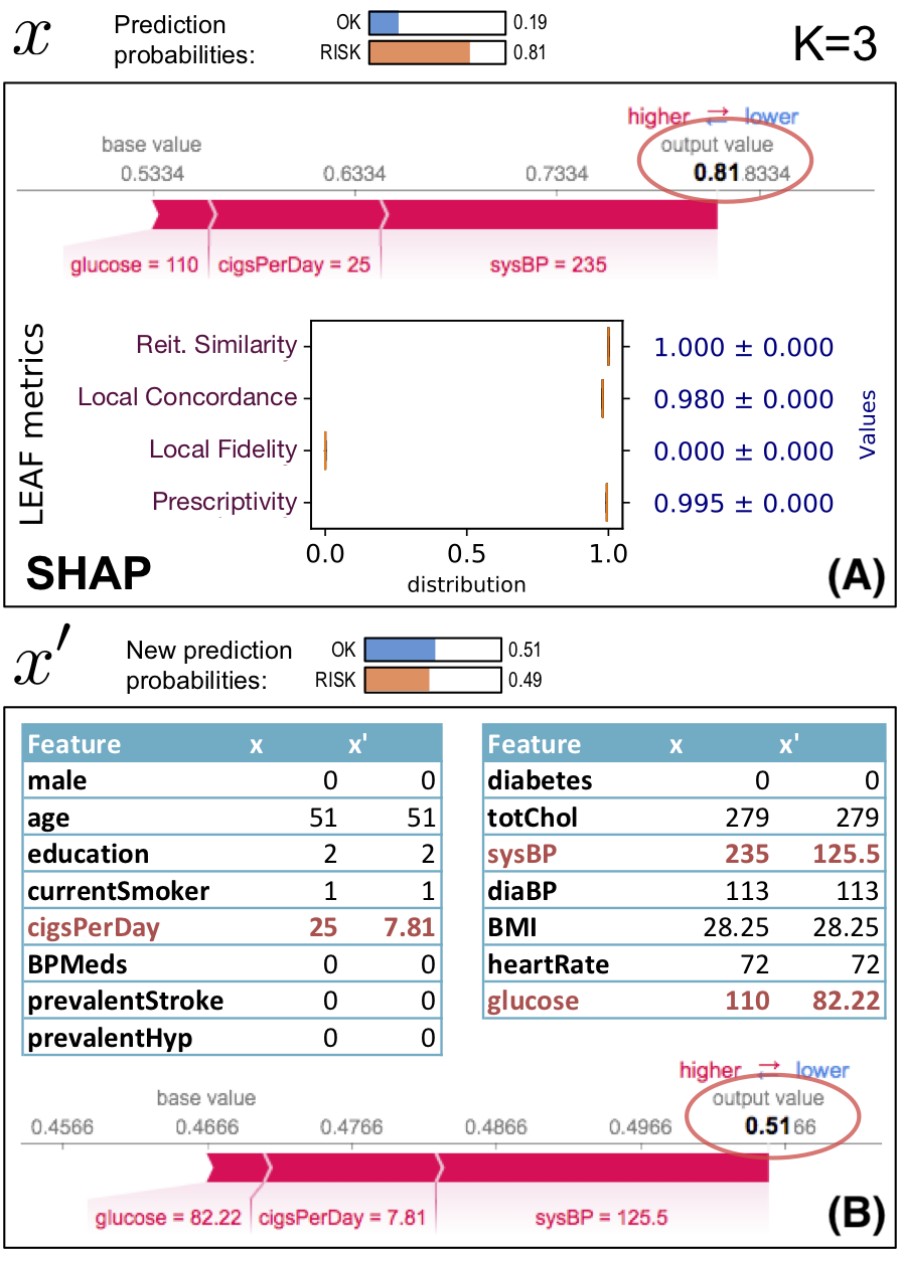

**Figure 6** Prescriptive use of an explanation.

- *What classifier should be trained to have high accuracy in the LLE models?* LIME has low local concordance for some classifiers (kn, mlp, svc), even for high values of $K$. SHAP instead shows increasing concordance levels at the increase of $K$, for all classifiers. Therefore SHAP is a better choice for local concordance, unless the black-box classifier is a linear or a logistic one.
- *What choices should be made to have explanations with high local fidelity?* Surprisingly, high local fidelity explanations can only be achieved by using some classifier categories

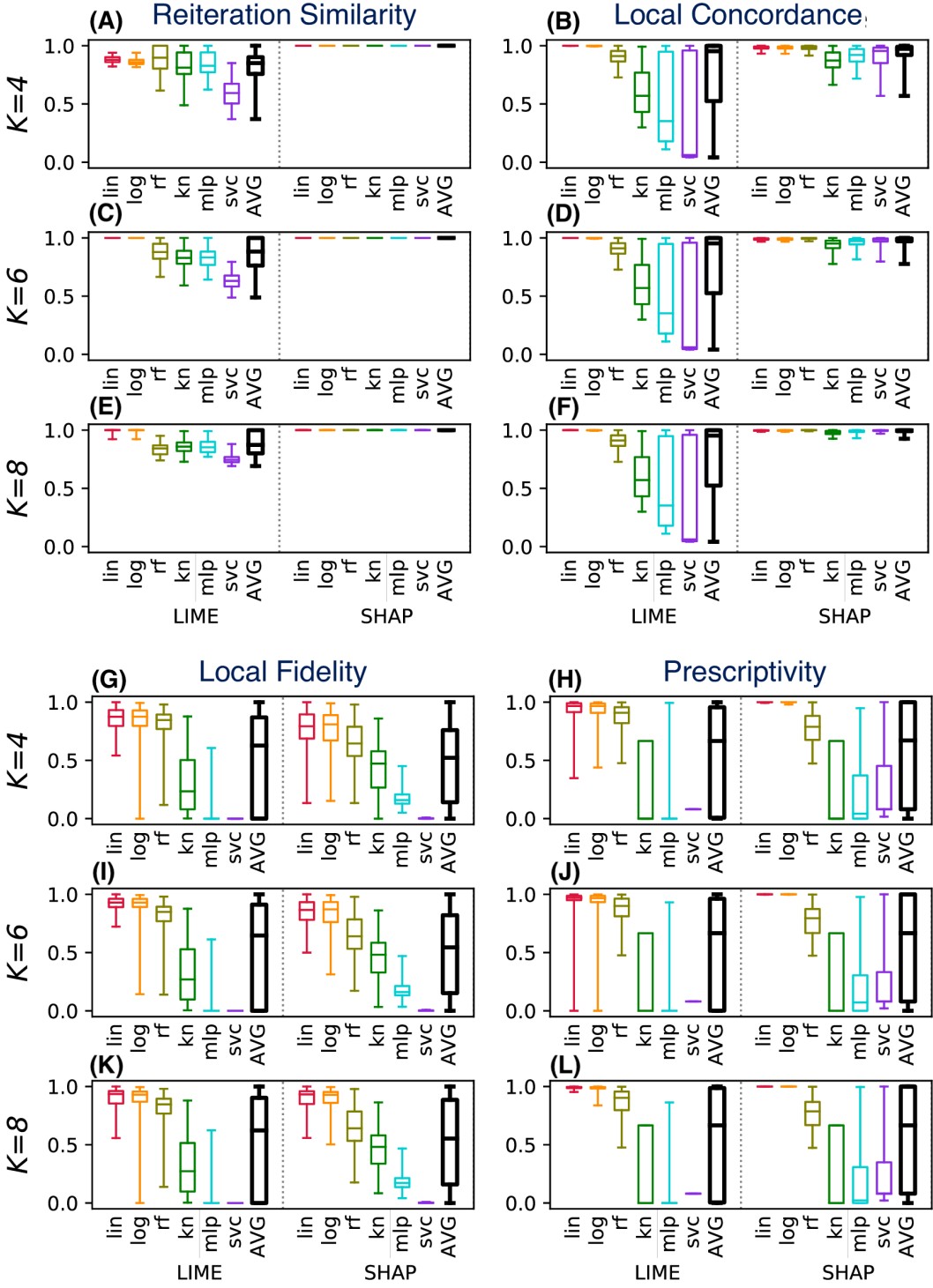

**Figure 7** **(A–L) LEAF used to support the decision of the best explainable classifier (case P2).**

(linear, logistic, and random forest for LIME). Increasing the conciseness does not appear to increase the local fidelity significantly, at least in the tested range of $K$. Some classifier categories (mlp, svc) show very poor explanation fidelities, regardless of the explainability method used.

- *What choices should be made to have prescriptive explanations?* Again, explanations can be used in a prescriptive way only for some classifier categories (lin, log). SHAP appears to have high prescriptive power even for low values of $K$, but only for a few classifier categories (lin, log and moderately for rf). LIME requires higher values of $K$ than SHAP to generate prescriptive explanations consistently. Other classifier categories (kn, mlp, svc) have poorly prescriptive LLE models, independently of the value of $K$ and the method used.

To summarise the example in Fig. 7, SHAP should be preferred because it shows a higher reiteration similarity and local concordance. It also shows good prescriptive power, at least for some classifiers. LIME should be preferred only if the user wants a higher local fidelity with low values of $K$.

Since the *drug* dataset used in Fig. 7 has only $F = 10$ features, SHAP exhibits a deterministic behaviour and perfect reiteration similarity, local concordance and prescriptivity are more favorable for SHAP. Figure 8 shows the results for the same metrics on the other three considered datasets (with $F = 279$, 15 and 30 respectively), computed using the same experimental settings of Fig. 7. For these datasets, SHAP does not use the deterministic algorithm, resulting in unstable explanations (B), with limited local concordance (D).

These observations lead us to conclude that SHAP is not more stable than LIME in the general case, and the reported advantage of Shapley values can only be exploited in practice for datasets with few features.

## DISCUSSION

Modern ML techniques are applied to an ever-growing number of domains, but often these ML models are intrinsically unable to provide explanations of their inner decision processes. XAI algorithms tackle this problem by enriching black-box labels with human-understandable explanations in the form of decision trees, rules or linear approximations. Many explainability methods have been proposed over the past years, and the necessity of comparing and evaluating the different explanations they produce has emerged. Several metrics have been proposed in literature for this task; however, when focusing on local linear explanations, there are no quantitative and systematic ways to assess and compare LLEs. We argue that a toolkit for explanation assessment is fundamental for the adoption of XAI algorithms and, as a consequence, of black-box ML models.

In this paper, we propose a general framework that we believe can be of great interest to most users, practitioners or researchers struggling with interpreting machine learning models via *post-hoc* explanations. It is worth stressing that the results of our work are not intended to identify the best explainable approach that fits every scenario, but rather to provide a well-grounded framework to provide such assessment case by case. In this

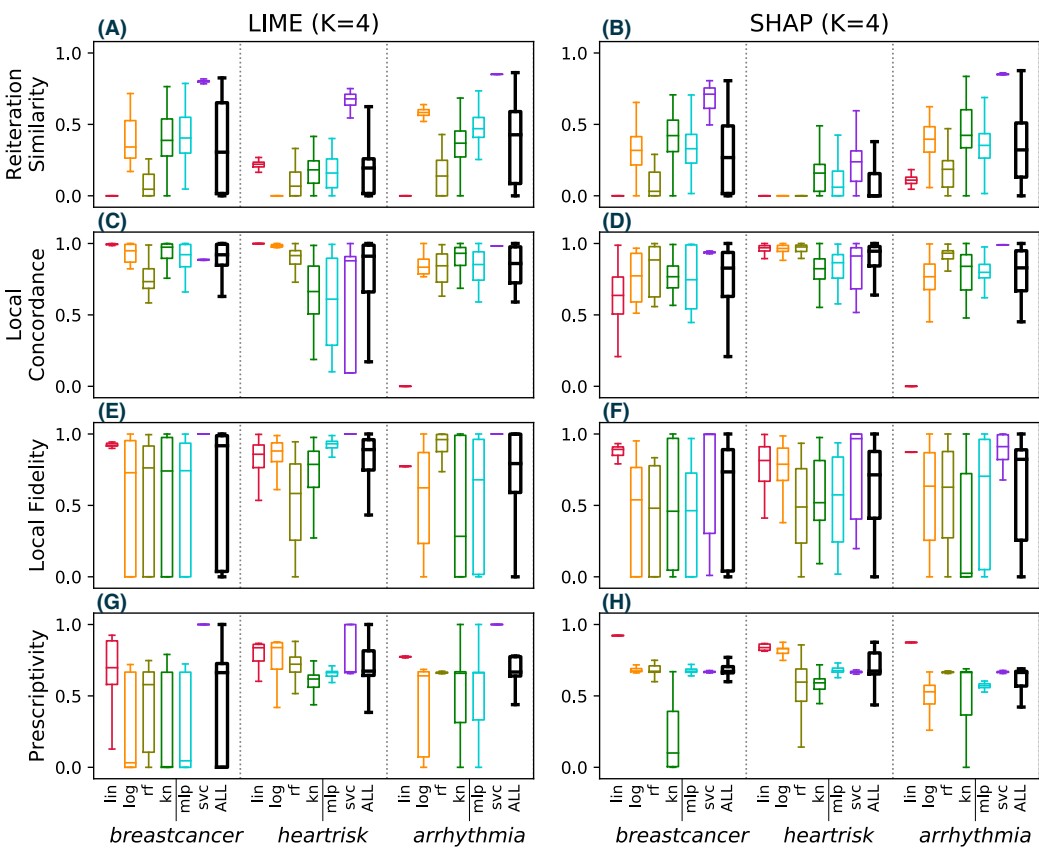

**Figure 8** (A–O) Observed metrics values across different classification tasks.

direction, we decided to not be constrained to a specific prediction task, but rather to test LEAF on different datasets and models, to fully describe the potential of a systematic approach to evaluate explanations. Our detailed experiments show that even widely adopted methods such as LIME and SHAP are prone to lack of reiteration similarity, low conciseness or even provide insufficient explanations for the correct label (i.e., the *K* selected features are not enough to categorise the datapoint). This supports our claim that explanations should always be paired with quality metrics to ensure they meet the requirements set by the user.

One of the major limitations of the present study is that we mainly focused on LIME and SHAP - however, we argue these are the two state-of-the-art model-agnostic XAI techniques for Local Linear Explanations. However, the described approach can be extended to analyse other explainability methods as well, both model-aware and model-agnostic ones, as long as the provided explanations are LLEs. Moreover, we restricted our attention to tabular data only, since less structured data are typically converted into tabular form to build interpretable models.

We argue that further analyses of the introduced metrics would be an interesting direction for future work. For instance, the idea of *reiteration similarity* has been developed

as a comparison between sets of features, without taking into account their relative importance, but if the rank of the explanatory features is considered particularly relevant for a specific domain, a weighted reiteration similarity can be defined. Moreover, reiteration similarity is shown as one of the major weaknesses of the analysed XAI methods. In particular, explanations of some classifier categories, like neural networks, seem to be plagued by unstable explanations. In our experiments, we observed that the LIME method is highly affected by unstable explanations and low local concordance, and SHAP deterministic behaviour is in practice limited to simple datasets only. We believe that the lack of a reiteration similarity control in XAI tools is a major obstacle in their adoption. Future works should target the reiteration similarity metric to built novel ensemble approaches, e.g., providing bagged/boosted explanations, and/or design explanations methods that provide some form of statistical accuracy/precision. Moreover, more complex refinements of this metric could be defined, as already mentioned in 'Methodology', to target additional aspects like feature rankings [66] or feature weights.

Local fidelity as defined in *Ribeiro, Singh & Guestrin (2016c)* also shows some limitations: the synthetic neighbourhood generates potentially out-of-sample data points which could lead to unreliable black-box model behaviours, influencing the overall explanation quality. The impact of a local fidelity score with a different sampling distribution is an important research direction that has not been fully addressed in this work.

We also believe that explanations interpreted as prescriptive control strategies could become an important source of information for practical action planning, as illustrated by the hearth risk example. To the best of our knowledge, such usage of LLE models has not been fully exploited so far. Measuring the effectiveness of a LLE to reach the decision boundary is a crucial part of any prescriptive strategy, and this is captured by the *prescriptivity* metric. Future work in this direction should involve the concept of *actionable* features, i.e., only consider changes on controllable features inside a constrained range, as well as measuring some form of reiteration similarity on the boundary or considering a trajectory instead of a single boundary point, similarly to individual recourses (*Joshi et al., 2019*).

It is worth noting that the actual prescriptivity definition could indicate out-of-distribution $x'$, which have limited to no practical value. Reducing the feature space to an *actionable* space could therefore constrain the problem to avoid generating inconsistent $x'$. Moreover, we acknowledge that the blind interpretation of the prescribed sample $x'$ should not be considered in a causal way, since the target only flips the black-box prediction. For instance, in Fig. 2 the prescribed change in the highlighted features does not automatically imply that the patient will heal from her underlying health conditions, but just that the modified behaviours and features flip the classification in the decision support system.

A first, organic direction for future work is the inclusion in LEAF of other metrics for explanation evaluation that have been recently introduced in literature. For instance, in *Alvarez-Melis & Jaakkola (2018)* the *faithfulness* metric is introduced, with the aim of testing if the relevance scores are indicative of "true" importance. A second candidate is the *Area Over the Perturbation Curve* (*Samek et al., 2017*), as it has been shown to be valuable for assessing explanation performances of saliency maps but could be generalized

for dealing with tabular data. Moreover, it would be interesting to integrate LEAF with the *RemOve And Retrain* (*Hooker et al., 2019*) iterative process, as it provides an empirical (albeit computationally expensive) method to approximate the actual relevance of the features selected by an explanation method. In principle this could be used to improve the interactive feedback loop proposed for the (**P2**) scenario. Another direction for future work is to complement the metrics of LEAF framework with a set of sanity checks based on measures of statistical reliability, as suggested in *Tomsett et al. (2020)* for saliency metrics.

## CONCLUSIONS

In this paper, we focused on the metrics that can be applied to model-agnostic explainability methods generating Local Linear Explanations, reviewing both existing metrics (interpretability, local fidelity), clearing ambiguities in the existing definitions (local concordance), and defining new ones that look at critical key points of the explanation process (reiteration similarity). Given the growing interest in using explanations to support decision systems, we also defined a general metric (prescriptivity) that evaluates how much a Local Linear Explanation can be used to change the outcome of a black-box classifier.

Our Python library, named LEAF, implements the aforementioned evaluation metrics for LLEs. We described how LEAF can be helpful in tackling two common scenarios, namely the *explanation of a single decision* (**P1**) and the *model development* (**P2**) use cases. An extensive set of experiments, using LIME and SHAP on four datasets, shows how the proposed metrics are fundamental to assess the trustability of LLE explanations.

We propose LEAF as an open framework, where researchers are encouraged to include additional metrics and benchmarks to foster the rigorous inspection and assessment of local linear explanation methods for black-box ML models.

### Funding
This work was supported by the Regional Research Project ''Casa Nel Parco'' (POR FESR 14/20 - CANP - Cod. 320 - 16 - Piattaforma Tecnologica ''Salute e Benessere'') funded by Regione Piemonte in the context of the Regional Platform on Health and Well-being and from Intesa Sanpaolo Innovation Center. The funders had no role in study design, data collection and analysis, decision to publish, or preparation of the manuscript.

### Grant Disclosures
The following grant information was disclosed by the authors:
Regione Piemonte.
Intesa Sanpaolo Innovation Center.

### Competing Interests
The authors declare there are no competing interests.

## Author Contributions

- Elvio Amparore conceived and designed the experiments, performed the experiments, analyzed the data, performed the computation work, prepared figures and/or tables, authored or reviewed drafts of the paper, and approved the final draft.
- Alan Perotti conceived and designed the experiments, analyzed the data, authored or reviewed drafts of the paper, and approved the final draft.
- Paolo Bajardi conceived and designed the experiments, analyzed the data, prepared figures and/or tables, authored or reviewed drafts of the paper, and approved the final draft.

## Data Availability

The source code and data is available at GitHub: https://github.com/amparore/leaf.

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
