# Peer review of "To trust or not to trust an explanation: using LEAF to evaluate local linear XAI methods"

_PeerJ Computer Science, doi:10.7717/peerj-cs.479_

## Round 0.1 · original submission · Major Revisions

Thank you again for your high-quality submission to PeerJ CS. Although the reviewers' suggestions, questions, and comments, taken individually, may not require a major change or a large amount of additional work, their sheer number warrants a "Major Revision".

Reviewer 2 has a large number of questions and comments, and excellent suggestions for improving the paper, particularly regarding the discussion of related work and an additional experiment with neural networks of increasing size. All of these should be addressed.

Please also consider the other two reviewers' comments, particularly Reviewer 1's comment regarding the definition of prescriptivity and Reviewer 3's comment regarding deeper networks.

It would also be useful to try to address Reviewer 3's request for more theory, but this is not a requirement for acceptance from my point of view.

·

Basic reporting

Generally, there is no basic issue. English language is appropriately used throughout. Introduction, context and references are reasonably clear and straightforward. Structure and formatting seem reasonable. Figures are well displayed and clear-cut enough. At least one of the dataset is directly provided in the github link; the others are referenced.

Experimental design

The four metrics chosen appear to be well-motivated and within the scope.
Research question is well-defined. The paper simply provides a set of metrics for users to compare explainability.

The definitions of the metrics are straightforward enough. There is complex processing, transformation or manipulation that will become real impedance towards the rigor of the investigation.
1. Fidelity is potentially overlapping with local concordance. However, by pointing out the instability of fidelity due to sampling, the author has been able to demonstrate the difference.
2. Possibly minor issue: 1/2 is used in the definition of prescriptivity. In many contexts, thresholding is not strictly 1/2. The presecriptivity is describe w.r.t model f that outputs a scalar value rather than vector (hence so does the explanation model). This should not be a problem since generalization is probably easy to do.

Replication should be possible, given the link to the code.

Validity of the findings

The results appear valid, as in the graphs showing metrics are plotted as the metrics are defined. No real complex processing to complicate the validity. The main results seem to be direct comparison and observations of the resulting values of respective metrics.

Possibly important insights are present, especially the fact that different explanation methods such as SHAP and LIME do not agree with each other.

Underlying data have been used in previous researches, and likely to be reliable.

Conclusion is concise and clear.

·

Basic reporting

The paper is well written, easy to follow and understand, and appropriately structured. The introduction and background sections provide suitable context for the work, and are generally well-referenced, incorporating several review papers in the area in addition to further useful contextual references and the key papers for the methods used. However, some important prior work specifically on metrics for explanations has been missed (details below). The figures are high quality and appropriate for illustrating the ideas and work presented, though some of the captions could be expanded a little (details below). The paper is self-contained and presents a suitable selection of results to demonstrate the utility of the explanation metrics.

Specific comments (line numbers refer to lines as labelled in the supplied paper draft):

- Line 30: the word “symbolic” is not necessary here – it could potentially add confusion for the reader with specific preconceptions about symbolic AI. I would recommend removing it and leaving “human understandable clarification” (and similarly where this term is used elsewhere).

- Line 76: there is a typo – “where” should be “were”

- Line 115: N(x) has not been introduced at this point in the paper, so should either be replaced with text describing it, or the definition of N(x) should be moved into this section

- Line 120-121: in the equation for LIME’s loss, I think the summation should be over z,z’ in N(x) (the neighbourhood of x), whereas the notation for the normal distribution has been used by mistake. Additionally, z’ should be defined explicitly as the “interpretable representation” per the original LIME paper – it’s a bit unclear what z’ means here and so what the pair z,z’ represents for the unfamiliar reader

- Lines 121-122: it is worth pointing out where this detail comes from as it isn’t, as far as I can tell, in the main text of the original LIME paper. The specific kernel I believe is the default in the LIME source code, but this kernel doesn’t have to be used (and indeed the choice of kernel can significantly alter results).

- Line 131: mu should be defined explicitly in the text

- Overall sections 2.2 and 2.3 would benefit from some additional verbal description to help the reader interpret the equations. It is difficult to form an intuitive understanding of the equations’ meanings without some prior knowledge of these kinds of models, and a small amount of additional text description would help the more general reader. Additionally, some reference to the variants of the SHAP algorithm and mention of which variant was used in the current study should be added, with some explanation about the possible implications for the results.

- Line 149-151: I think this needs a bit of clarification – presumably it is the function f(x) that does not vary much in the region around x (for a feature assigned low importance)? At the moment it reads like x itself does not vary much in the neighbourhood of x, which doesn’t make sense to me… This section is good, though, and makes an important point about the interpretation of these different methods, despite them both being in the same family of techniques!

- Secton 3, starting line 177: I have some concerns about the terminology used to name the metrics. The definition of interpretability as pertaining to human understanding (per Guidotti et al. 2018) is fine, but I don’t think it’s appropriate to use “interpretability” as a name for a metric that only measures one small element of this (the explanation length), especially when the broader field of “interpretable machine learning” studies a whole range of different facets under the term “interpretability.” I strongly suggest renaming this metric to something less broad and more precise, and replacing the use of “interpretability” with a more suitable name throughout the paper. Line 188: similarly, “fidelity” has been used to describe the general property of the correctness of an explanation in other interpretable machine learning literature, rather than as the name of a specific metric assessing one aspect of this (e.g. the “faithfulness” metric in https://arxiv.org/abs/1806.07538 or Area Over Perturbation Curve, AOPC, in https://ieeexplore.ieee.org/abstract/document/7552539 ). Perhaps a more suitable name might be “local fidelity”? Finally, “stability” has been used to refer to explanations that change very little in response to small perturbations to the input – again in, for example, https://arxiv.org/abs/1806.07538 . This is a bit different from the author’s proposed metric, so perhaps a different name is warranted? Apologies if these requests seem a little pedantic – but I think it is important to try to maintain some consistency in vocabulary within the field.

- Line 192: were these instances from the training set or test set? Do the metrics report different distributions of values between training and test set instances?

- Paragraph starting after line 232, before equation 4: there is a superfluous “element of” symbol

- Figure 1: the figure could use a little more explanation in the caption, especially regarding the meaning of the arrows

- Figure 3: make explicit in figure legend that this is for the *same* input data point and same ML model – and add some explanation of why the output value is different (0.72 vs 0.68) for the two explanations (presumably due to using K=4 rather than the full feature set?) This figure also needs a little more description of the numbers in the visuals as, these are the default outputs of the methods and differ a bit in their presentation. This could be added in the main text or in the caption.

- Line 242: the comment on the reason behind LIME and SHAP’s popularity seems like speculation, so should be noted as such (other possibilities include that they both provided user-friendly Python implementations, provide visualisations, and have been incorporated into some companies’ XAI solutions).

- Line 245: Point of clarification - SHAP can be used on image classifiers without using superpixels as the interpretable representation. The mapping to an “interpretable representation” in this case is just the presence/absence of the original pixels, rather than of superpixels.

- Line 315: according to figure 5, the “not at risk” boundary should be 0.49, not 0.51

- Line 381: (also in the abstract) “even explain the wrong label” – what does this mean? I don’t think this is properly explained in the text (apologies if I’ve missed an explanation somewhere).

Experimental design

The experiments chosen to illustrate the utility of the proposed metrics and toolbox are appropriate and quite thorough. There’s always more that could be explored but I think the authors have chosen a useful set of experiments to illustrate the key concepts. They have used freely available datasets and tools so that their experiments can be reproduced, and no private data has been used inappropriately. As far as I am aware, these datasets have been cleared for open use and no ethical issues have been raised with any of them previously. I have a couple of comments and requests for some clarifications:

- Line 241, Section 4: we need more details about the ML models: what was the train/test split? What was the test performance of each model? How likely was each model to over-fit? Were the 100 instances presented in the figures all classified correctly? If not, was there a correlation between incorrect classification and stability/any of the other metrics? Would also be good to briefly mention the tooling used (scikit-learn?), with reference to the github repository)

- Line 260: the same number of samples were used in SHAP and LIME – can authors elaborate on the equivalence here? Given the methods work differently, does this make sense? It’s OK if this is an arbitrary decision, but should be noted as such if the equivalence is not really appropriate.

- Results showing very unstable explanations for NNs/SVMs: this is important and should be expanded on. It seems likely that the models are fitting a highly nonlinear boundary and this is affecting the stability results. However, it’s difficult to generalise the results given only a single NN architecture (MLP with 100 hidden neurons) was tested. A simple experiment that would help illustrate the behaviour of the stability metric with nonlinear models would be to measure it for NNs with increasing numbers of hidden neurons (starting from just 1, also monitoring model performance). I would request the authors perform this experiment for one or two datasets, as this would significantly improve the discussion of the stability metric and aid reader’s intuitions about its behaviour on more complex models.

Validity of the findings

The findings are a demonstration of the behaviour of the proposed metrics using a suitable variety of ML models and datasets. I have asked for a further experiment to expand on the findings for NNs in the previous section of the review. The conclusions are sound and well written. I have a few specific questions/comments:

- Line 279: results indicating prescriptivity > fidelity – are the scales of these metrics comparable like this, so that you can be confident saying the behaviour of the local explanation models is better at the boundary than locally to the data point? How does this change between points classified with high confidence and points classified with low confidence?

- Regarding the local neighbourhood of an input, N(x): the relevance of the local “fidelity” score is going to depend strongly on whether the local sampling method samples data points that are within the distribution of the original training data, or produces samples that are out of distribution. The authors note that the sampling method will affect the fidelity metric (potentially favouring LIME) but do not expand on this sufficiently in the discussion – especially the potential impact of generated samples being out of distribution and what this means for interpreting this metric.

- The authors note the limitation of their proposed stability metric – that it does not take into account feature ranking – but I think this is more serious than implied: it seems a severe limitation, especially for models with fewer features. It is possible that an explanation method could produce explanations with entirely opposite feature importance rankings yet receive a high stability score. This at least requires further discussion and should be highlighted more prominently when the metric is introduced. However, the metric as-is does reveal some issues with the explanation methods tested, so it is at least useful in its current form. A possible improvement to this metric might be to use something like Rank-Biased Overlap: http://dx.doi.org/10.1145/1852102.1852106 rather than Jaccard similarity – perhaps mention this in the discussion?

- The discussion of “prescriptivity” should include some mention of causality and the possibility of mis-interpreting results with predictive rather than causal models. The equation to generate the contrastive input also has the potential to create out-of-distribution data, and this should be noted and the implications discussed.

- Some metrics have already been proposed for XAI methods, for example Area Over the Perturbation Curve (AOPC) and “faithfulness” mentioned above. Though not for LLE methods specifically, these metrics should at least be mentioned as prior work. Even better would be to indicate how these metrics differ, or are similar to, the metrics presented in this paper.

Additional comments

Overall this is a useful and well-written paper addressing an important and under-studied topic in XAI. The provision of a Python toolbox implementing the proposed metrics is especially welcome. One final point: there has been some work in the area of metrics previously that is not mentioned by the authors, perhaps as it has focused on image classifiers/DNNs and saliency map explanations, rather than LLEs specifically (see mentions of AOPC and “faithfulness” previously; also the ROAR method: https://arxiv.org/abs/1806.10758 ). However, these papers make transferable points about metrics so their inclusion in the discussion/prior work seems important. I am aware that it is rather uncouth of me to suggest referencing my own work, but our paper “Sanity Checks for Saliency Metrics” https://arxiv.org/abs/1912.01451v1 seems particularly appropriate here as it proposes methods to assess the statistical validity of explanation metrics. I would not expect the authors to implement these checks in the current paper, but it should certainly be considered as further work (or, alternatively, an explanation of why the proposed sanity checks would not be appropriate should be given).

Reviewer 3 ·

Basic reporting

This paper presents a set of metrics for the evaluation of local linear explanations. In principle, evaluating the explainability of local linear models is an important problem. Unfortunately, it is hard to figure out the novelty of the proposed method in the currently submitted form.

Experimental design

- The authors use medical datasets: drug consumption, arrhythmia, breast cancer and so on.
- The authors define problem1 (P1) and problem (P2), and answer the questions using proposed method
- The authors define four metrics to evaluate local linear explanation, but the definition looks heuristic.
- The authors describe detailed information of four proposed metrics.

Validity of the findings

This paper suggests a new framework “LEAF”(Local Explanation evAluation Framework) to evaluate the local linear explanations. LEAF consists of four metrics so that users can consider both Explaining single decision and Model development. LEAF reveals that either LIME or SHAP cannot ensure its robustness for every scenario, and suggests users apply such frameworks case by case.

Additional comments

Here are comments to authors. It would be good to revise the draft to improve the paper especially on weakness.

Strength
- The authors define key questions about XAI methods, as well as answer the questions using the proposed method. It looks well organized.
- This paper shows overall good presentations and its figures are easy to understand.
- LEAF can evaluate such existing frameworks as LIME, SHAP. Also, LEAF seems to be flexible and scalable for various models and datasets.

Weakness
- LEAF looks somewhat heuristic. Suggested four metrics (fidelity, local concordance, stability, prescriptivity) are defined without theoretical proof. The authors only mentioned the reason why the four metrics are reasonable. It would be better to add more theoretical approaches.
- As the authors mentioned in the Discussion, LEAF is applicable to local linear explanation methods. Though the authors raised universal questions about XAI, the suggested method deals with only two local linear explanation methods.
- The authors did not use complicated deep learning models in their experiments. Deep learning has been widely used in many areas, but they only used a simple perceptron structure.

---

## Round 0.2 · Minor Revisions

Please address Reviewer 2's comments as well as the typos, suggested wording changes, questions, and comments listed below.

"is a scarce consensus" -> "is no consensus"

"regards the decision process" -> "concern the decision process"

"Local Linear Explainers basic concepts" -> "basic concepts concerning Local Linear Explainers"

"evaluating the explanations provides" -> "evaluating the explanations provided"

"like for" -> "e.g., for"

"i.e. the" -> "i.e., the" (and in other places where i.e. is used)

"with form" -> "of the form"

"Shapley-like used" -> "Shapley-like values used" ?

What is the meaning of "input dataset" in Section 2.1 and how does it affect the standard deviation used in Equation 2? Surely, it is not the global standard deviation of a feature in the training data, for example.

The last sentence of Section 2.2 repeats information provided earlier in the section.

"In principle the" -> "In principle, the"

"The more important is a player" -> "The more important a player is"

"resuinting"

"An important observation is that SHAP is deterministic only in its theoretical formulation" -- Incorrect! TreeShap is deterministic!

"However SHAP limits" -> "However, the official SHAP implementation limits" (give a link to the implementation in a footnote)

"For SHAP importance" -> "For SHAP, importance"

"be an intrinsically black-box" -> "be an intrinsically black-box model"

"used (conciseness, local fidelity)," -> "used ones (conciseness, local fidelity),"

"In LIME the" -> "In LIME, the"

"an information criterion (like the Akaike criterion or the Bayesian criterion) to get the most relevant," -- do you have a reference for how to do this?

"usually measured using the F1 score" -- do you have a reference for this?

"Fidelity", "Concordance" -- I would not capitalize any of these.

"By using the N(x)" -> "By using the neighbourhood N(x)"

"relationship with the conciseness" -> "relationship with conciseness"

"it also consider"

"also deterministic methods like SHAP may fail" -> "methods like SHAP may also fail"

"could not converge to a unique solution" -> "may not ..." or "cannot ..." as appropriate

"how effective is a LLE when" -> "how effective a LLE is when"

"is on the extreme side of the classification." --- what does this mean?

"to LIME and SHAP popularity" -> "the popularity of LIME and SHAP"

"superpixels/pixels patches" -> "superpixel/pixel patches"

"R = 50 explanations." -- It would be good to clarify that this does not refer to the number of trees in the forest in this context.

"Only the first explanation is shown, even if LIME/SHAP compute 50 explanations each to get the metrics." -- "... even though LIME/SHAP are run 50 times to obtain 50 explanations so that the metrics, such as reiteration similarity, can be computed." (the wording here is quite misleading!)

"SHAP also shows" -> "In contrast, SHAP shows" ?

"0.762 and 0.375" -> "it is 0.762 for LIME and only 0.375 for SHAP"

"Evaluate the ...", "Select the ..." in the section titles: I would use "Evaluating the ...", "Selecting the ..." instead, rather than commands

"kn (k-neighbors with 3 clusters);" -- really? Clusters, not neighbors?

"non trivial pattern" -> "non-trivial pattern"

Figure 8 does not show local fidelity! Explain why.

"are currently applied" -> "are applied"

"In this paper we propose" -> "In this paper, we propose"

"both model-aware and model-agnostic" -> "both model-aware and model-agnostic ones"

"converted in" -> "converted into"

"between set of" -> "between sets of"

"one of the major weakness" -> "one of the major weaknesses"

"rankings Webber et al."

Paragraph break before "It is worth noting..."

"constraint the problem" -> "constrain the problem"

"For instance in" -> "For instance, in"

"prescripted change" -> "prescribed change" ?

"in model development" -> "in the model development"

"like the faithfulness" -> "like faithfulness"

"In this paper we focused" -> "In this paper, we focused"

"and defining new ones" -> ", and defining new ones"

"to revert" -> "to change" ???

"explaination"

Bibliography: "in r" -> "in R", "ai" -> "AI" (more than once)

·

Basic reporting

Many changes have been made to enhance the clarity of the text, addressing the valuable comments from reviewer 2.

Experimental design

As before, there is no complex processing that will need detailed technical attention (apologies for the typos in the previous review).

Validity of the findings

As before, the findings as they are do shed light into the behaviors of some methods tested.

Additional comments

The paper may contribute well towards the XAI community by providing more motivations to define the scope of explainability.

·

Basic reporting

All comments addressed

Experimental design

All comments addressed - great to see the new neural network results!

Validity of the findings

All comments addressed

Additional comments

Thanks very much for taking the time to address the points raised in the reviews! All my previous points have been addressed very well.

The only minor point I would raise now concerns the new final paragraphs in Section 5 (Discussion), added to address my point about citing prior metric work in this area. I think the metrics mentioned - "faithfulness" (Alvarez-Melis and Jaakola 2018) and AOPC (Samek et al. 2017) - are not quite as specific to particular settings as the authors suggest. Faithfulness does not have to be used in a self-explaining loop - see e.g. the IBM AIX360 toolbox for a generic implementation: https://aix360.readthedocs.io/en/latest/metrics.html . Similarly AOPC does not necessarily have to be used for images & saliency map explanations - its definition is generic, though it has usually been applied to images/saliency maps. Finally, the reference Tomsett et al. 2020 is about the evaluation of the statistical reliability of explanation metrics, rather than describing any metrics itself.

This section just needs a small edit to clarify these points and ensure the references are incorporated correctly. The point about other metrics is not necessarily that they could be added to the toolbox as further work, but that these represent previous efforts at creating explanation metrics and so should be cited for completeness. The point in Tomsett et al. 2020 is that any proposed metrics for explanations need assessing to evaluate their statistical reliability - though this can certainly be left to future work.

---

## Round 0.3 · accepted · Accept

Thank you for incorporating the last round of comments.

If you still have the option to make small changes once you have received the proofing PDF (not sure about this): "explanations ... concerns the decision process" should be "explanations ... concern the decision making process" (first page).